# Physical and Metrological Approach for Feature’s Definition and Selection in Condition Monitoring [note 1]

**DOI:** 10.3390/s19235186

**Published:** 2019-11-26

**Authors:** Giulio D’Emilia, Antonella Gaspari, Emanuela Natale

**Affiliations:** Department of Industrial and information Engineering and of Economics, University of L’Aquila, 67100 L’Aquila, Italy; antonella.gaspari@univaq.it (A.G.); EMANUELA.NATALE@UNIVAQ.IT (E.N.)

**Keywords:** condition monitoring, accelerometer, laser vibrometer, system model, feature selection, ANN, classification accuracy

## Abstract

In this paper, a methodology is described aiming at emphasizing physical and metrological criteria in feature selection for condition monitoring of a real scale mechatronic system. The device is used for packaging applications according to the movements of its end effector, driven by a couple of brushless servomotors and a kinematic mechanical linkage. The approach is hybrid, meaning that the starting feature set is built with reference to both experimental data from different sensors and to the indication of a simplified kinematic and dynamic model of the mechanical linkage itself. A critical comparison and mixing of theoretical and experimental data, based also on a physical interpretation of differences, suggests some more features, with respect to the classical ones, of hybrid type, which could be mostly correlated to the effects of statuses and defects of the system to be identified. The whole procedure is step by step validated, in order to evaluate the variability of features, throughout the whole procedure. The variability is analyzed depending on the actions that are realized in order to define, select, and use the proposed features for data processing by advanced algorithms, like the most typically used classifiers and artificial neural networks. A comparison with the state-of-the-art automatic feature’s selection procedure is also presented. Experimental results show that the proposed methodology is able to classify with high accuracy many statuses of the mechatronic system, which are only slightly different as for set-up settings and/or mechanical wear and lubrication conditions of mechanical parts of the mechatronic system. Issues to be pursued to a more effective generalization of the method are also discussed.

## 1. Introduction

Condition Monitoring (CM) of mechatronic systems in an industrial scenario is a practice more and more used to improve the use of the asset with reference to many types of machinery and operative situations [1,2].

Sometimes, specific tasks are of interest, like a suitable definition of the working parameters at the set-up of the system and/or identification of critical conditions due to fault and/or wear for smart maintenance [3,4,5,6,7], to now be realized in modern operating scenarios like Industry 4.0 or IoT [8].

If we consider the main requirements of CM, procedures should be selective and resolute, with reference to the status of interest, reliable, with reference to the capability of operating well also in difficult conditions from the environmental point of view, of general validity, in order to be transferable to systems different from each other, maintaining the diagnostic capability.

All these characteristics require that all the steps of the CM activity are optimized, considering the aspects connected to the measurement of data (sensor choice and positioning, data acquisition, measurement of experimental data, sensor fusion, …) to the data processing and synthesis, to give support to the decision-making.

A remarkable role for the success of the CM application is connected to the variation of specific features, defined to synthetize the phenomena of interest. The feature’s behavior depends on many factors, like, for instance, the quantity of interest, the type of sensor used and the way data from different sensors are merged [9,10], the part of the operating cycle taken into account and, of course, sensor positioning throughout the kinematic chain [1].

According to the literature [4,11,12,13,14], features in the time and/or frequency domain are available that are expected to give reliable and selective information about the setting or about the presence of typical defects in the mechatronic devices. 

Improving the meaning, the accuracy, the selectivity and the resolution of features, especially in complex mechatronic systems, is not a trivial task. In fact, both huge amounts of data deriving from networks of sensors have to be merged and advanced procedures of data mining and processing should be used. Finally, they should be easy and light to calculate with reference to the data processing [15,16].

In a previous work [17], the authors aim at improving the classification capability of features, by using an integrated approach of different contributions. Sensor fusion has been used to have experimental information at different positions of the mechanism and to merge their experimental information to the indications of a simulation model of the kinematic and dynamic behavior of the system. This approach allowed for defining a wide set of features, able to take into account most of the phenomena of interest for the CM itself. The hybrid approach already proved to be very effective in diagnosis and prognosis for smart maintenance [18].

Addressing correctly the CM methodology towards a reliable, accurate, selective, and resolute classification of the status of interest asks for some requirements. The features the CM is based on should be exhaustive with respect to the possible phenomena of interest, meaningful from a physical point of view, robust with respect to the noise, coherent between each other and limited as a number. Due to these many reasons, feature selection is a key point of the CM strategy. A first selection criterion of the features based on their repeatability was proposed [17]. The debate about these topics is very active, with reference to both general approach and specific applications, like, for instance, those related to tool CM in production machine [19,20,21,22] or computer vision applications [23]. How much these methods are of general validity is still an open point for CM of mechatronic systems.

In [24,25], authors provide a useful analysis of the main techniques for feature extraction and selection, for CM applications of industrial interest. They propose a set of algorithms which automatically extract and select features, independently of both the type of the specific applications of interest and the classifier used, as a basis for the tasks that the maintenance-on-condition requires.

Other examples exist [26], providing a wide survey on semi-supervised methods for feature selection: most of the available algorithms make use of a series of nested algorithms that must be adequately known and characterized, so that they can work properly and return fruitful results. This scenario appears inter-disciplinary due to the nature of the topics, ranging from electro-mechanical know-how, to informatics, automatics, statistics, and so on. Therefore, entrusting some “black boxes” called for understanding the causes of problems linked to the ordinary industrial operations, and identifying their solution still represents a limitation, especially when the complexity of the assets/system can make application of tools more difficult. In other cases, if a lack of knowledge of processes is encountered, for instance when the process is quite new, a multi-step feature selection procedure is realized to get a reliable monitoring of the process [27].

The goal of the present paper resides in the possibility of increasing the awareness of the phenomena underlying the monitored processes by taking under control the data flows at all the main steps of the procedure, thanks to an increased robustness due to a physical and metrological support; this is believed to increase the general validity of the available tools.

The paper is organized as follows: Section 2 describes the approach, which has been applied to a test bench taken as an example of device of general use in a generic industrial plant. The mechatronic device under test is an industrial system for automatic packaging, whose behavior is strongly influenced by the initial setting of the control parameters and by the lubrication status of the ball screw along the vertical axis. Feature extraction and selection serve as a preliminary step of a training and testing procedure of advanced algorithms, whose performance must be assessed clearly, i.e., highlighting the specific need of the CM application. Section 3 summarizes the main results that have been reached, by applying a physical and metrological approach, in comparison with automatic procedures commercially available. Conclusions and future developments are provided at the end of the paper.

## 2. Materials and Methods

### 2.1. Test Bench and Simulation Model

The methodology has been implemented with reference to an automatic system in real scale (Figure 1a), able to realize an alternate linear motion along a vertical axis by means of one ball screw. One motor manages the motion, so the movement of the shaft is a pure translation, with sinusoidal motion law at a frequency in the range up to 5 Hz and a stroke up to 150 mm [17].

The scheme of Figure 1b shows the measurement apparatuses, with reference to both internal sensors to the system, i.e., angular position and motor currents transducers, and to the external ones, i.e., a Laser Doppler Vibrometer (LDV), bandwidth 50 kHz, and a Micro Electro-Mechanical System (MEMS) tri-axis accelerometer, bandwidth 200 Hz.

The acquisition from external sensors is carried out by means of a Data Acquisition System (DAQ) by National Instruments (Austin, TX, USA), at a sampling rate of 1000 Hz. A synchronization between the external data acquisition and the PLC has been realized, by using a trigger signal from the PLC to the DAQ.

By way of example, Figure 2 shows time diagrams of quantities measured by both internal and external sensors to the PLC.

The main steps of the methodology are according to the following ones:Realization of a representative kinematic and dynamic model of the system of interest. The theoretical model developed for the system has been extensively described in [10].Realization of experiments, corresponding to different operating conditions.Multiple runs of the model, considering as input different quantities, theoretical or measured ones [17,28]. The possible inputs are:
Angular position of the motor axis,Angular velocity of the motor axis,Electric current at the driving servomotor,Linear acceleration of the ball screw shaft,Linear velocity of the ball screw shaft.Comparison between outputs of the model and data deriving from measurements (angular position or velocity of the motor axis from the encoder, electric current at the driving servomotor from the internal sensor, linear acceleration of the ball screw shaft from the accelerometer, linear velocity of the ball screw shaft from the LDV).Calculation and selection of the most suitable features for the jload setting [17] and the lubrication state identification. Jload is a parameter of the control system, representing the load inertia at the motor axis.Application of advanced data processing techniques for classification.

It has to be noticed that steps 3 and 4 of the procedure are useful to identify specific time windows for feature computation, where the real behavior differs from the theoretical model due to peculiar effects of motion and control: this comparison can suggest many possible hybrid features to be used in the next steps, based on both experimental and simulated data. As an example, in Figure 3, a comparison between simulated and measured acceleration signals is shown: it can be seen that the simulated acceleration, obtained using as an input of the model the real current, presents a peak in correspondence to the dead center, unlike real acceleration. This means that the motion law is correctly followed, but the absorption of current is maximum at the dead centers, in particular for incorrect setting of jload. For this reason, the difference between peaks of real and theoretical acceleration, for example, can be a significant hybrid feature for the correct setting of the system. These aspects have been deeply treated in previous application of the methodology [17,28].

Nevertheless, previous works highlighted the need for paying particular attention to the feature extraction and selection in order to realize a reliable, accurate and efficient approach for CM of automatic systems [17,28].

In order to get reliable criteria of comparison among different possible solutions for feature extraction and selection, a careful analysis of the features accuracy is carried out, as explained in Section 2.3. 

Several metrics for the evaluation of the performances of classifiers have been considered, identifying the most suitable ones, depending on their meaningfulness for CM (Section 2.7).

### 2.2. Definition of the Initial Set of Features

Different quantities have been measured, both internal to the PLC, like tracking deviation (TD) and current (Mcur) of the servomotor used for motion control and external ones with respect to the PLC, like vibrations in different points of the kinematic chain.

On the basis of these quantities, the initial group of features has been built by adding to the traditional features of measured quantities, other features, suggested by the simulation model of the mechanism. 

The features that are traditionally used for CM (RMS, crest factor, kurtosis, …, in the time domain; amplitude of power spectra, band power, envelope, …, in the frequency domain) [4,12,13,14,19,20,21,22,23,24,25], and that are considered in this work, are useful in most applications to maintain the relevant information about the process or tool conditions [4].

The kinematic model is asked to indicate the way of defining other meaningful features in different ways, e.g., measured data of specific quantities and during specific time windows of the cycle, sensitivity analysis of the system to the condition to be classified (setting and/or defect), mixing of measured and simulated data, …. Therefore, in addition to traditional features, specific features built for the analyzed application and hybrid features are also considered, obtained by mixing experimental and simulation data, as explained in Section 2.1. A set of 101 features has been defined by this procedure (SET 1).

Automatic definition of the initial set of features has been also taken into account for comparison purposes, with reference to the accuracy of classification. Using literature and commercially available features extraction methods on measured data [29,30,31,32], a second set was created, having 100 features (SET 2), the same size as SET 1. Many possible approaches for feature extraction could be used. The proposed method was intentionally compared with a method of large utilization and easy application, in order to give a comparison with an approach that does not require physical knowledge of the application. In particular, the automatic sparse filtering feature extraction method has been used in this work.

It has to be pointed out that features belonging to SET 1 have been evaluated considering up to 10 s of acquisition, while the ones of SET 2 refer to a total length of acquisition of 2.5 s. From a preliminary analysis, this is due to the automatic tool limitations, in terms of unsatisfactory results as far the accuracy of classification—i.e., the accuracy less than 60%, or in terms of long computing times.

### 2.3. Feature Selection

Feature selection aims at improving the performance of a CM procedure, increasing the accuracy of classification and, at the same time, reducing computation load and time. For these motivations, this step is crucial for the procedure.

Physical criterion means that there are physical reasons why the features could be more significant like, for instance, kinematic amplification of displacements and accelerations, dynamic or electrical effects depending on inertial forces in particular windows of the operation cycle, and control settings of the PLC controller. 

Metrological selection means to evaluate carefully the repeatability of features, with respect to the differences of the mean values of the same features, when the statuses to be classified are changed.

By way of example, Figure 4 shows the relative difference of some features calculated on the basis of TD, at different jload; features, whose differences of the mean values are significant with respect to the repeatability, are chosen, in particular, PRC98_TD. For clarity, PRC98_TD is the 98th percentile of the amplitude of TD. Similarly to TD features, there are some features calculated on the basis of the measured current of the servomotor, which result as particularly significant with respect to the lubrication and setting conditions, as described in [10]. It has to be pointed out that this approach is not trivial, due to the need for taking into account also the operating modus of the mechatronic system.

For comparison purposes, automatic methods for feature selection have been applied to both SET 1 and SET 2.

A sequential feature selection method has been chosen and used, which is commercially available and of general application. It computes the number of misclassified observations of a classification model [33,34]. In fact, this method refers to the optimization of a function or criterion, which is typically based—in turn—on the training of a classification model and the prediction of values using this model. The classification model is based on a discriminant function [35,36] that can be of different types; in our case, the following have been tested, being the working ones: ‘linear’, ‘diaglinear’, and ‘quadratic’. Figure 5 highlights the main steps of data processing and of training and testing procedures for ANN and classifiers. Algorithms for raw data post-processing refer to the need of organizing and managing the data provided by the test bench (e.g., to organize the time sequence of the acquired samples), which are extensively described in [37].

### 2.4. Design of Experiments

The ability of features to evaluate setting characteristics has been tested, by varying the jload parameter, and by realizing variable lubrication statuses of the ball screw, related to different quality levels of it. The test condition is sinusoidal motion law of the end effector of the kinematic linkage, cycle frequency equal to 3 Hz, which is a very challenging operating condition for the mechatronic system, being close to its maximum rate.

Nine classes should be classified, obtained by combining three lubrication conditions and three jload values (3 × 3 = 9 classes), in particular:jload (3.0 kg cm^2^, 5.5 kg cm^2^, 8.0 kg cm^2^),lubrication (inadequate lubricant, G0; minimum lubricant, G1; regular lubricant, G2),

The effect has been investigated of some influencing parameters, like:length of the acquisition interval,added noise to the experimental data,number of test repetitions.

### 2.5. Training and Testing Procedure for Classifiers and for ANN

In order to evaluate the performances of classifiers and ANN, different configurations have to be compared with regard to the training data set and to the testing procedure. Each configuration is named using the following form: SET i − js − Lk – Fw,(1)

where:
i stands for the name of the group of features, as above-mentioned (SET 1 and SET 2);j denotes the total length of acquisition used for training of classifiers/ANN, as one of the following options: 10 s, 15 s, 20 s, 30 s or 40 s, meaning that, considering the whole number of windows on which the features are calculated, the total temporal basis used for the training procedure, is referred to a total length of acquisition of different independent experimental tests of j s; it has to be pointed out that the time duration of acquisition has been set with reference to a range including a minimum number of operation cycles (5 s) and a maximum one related to the need of acquiring too many samples. In addition, 40 s has been considered a correct trade-off, for the latter requirement.k specifies the temporal interval of each independent experimental test, on which the features are evaluated (5 s or 10 s). The ratio j/k defines the number of independent experimental tests used for training purposes and, therefore, it also denotes the number of observations used for training;w gives us the idea of the number of features used for classification:
○101 refers to all of the features of SET 1,○21 refers to 21 features of SET 1, calculated on the basis of Mcur and TD, derived from applying the so-called physical selection method of features,○16 are the features of SET 1 referred to the 98th percentile of both measured and simulated quantities available, derived from applying the so-called physical-metrological method for feature selection;○other, depending on the specific situation, as explained in the following.


As already said, the feature selection by means of the use of physical and metrological criteria should be compared to some state-of-the-art automatic feature selection methods, in order to benchmark and locate the proposed method and to investigate its robustness, also from a statistical point of view. For this reason, one of the following has been added to the nomenclature, indicating that the feature selection method is automatic, and based on linear, diaglinear, or quadratic discriminant function, respectively:‘linear’,‘diaglinear’,‘quadratic’.

### 2.6. Advanced Algorithms for Classification

The classifiers taken into consideration can be divided into two typologies: an ANN classifier,classifiers based on algorithms different from ANN.

The ANN used for classification is a two-layer feed-forward network, composed of 10 hidden neurons. Sigmoid transfer functions are used in the hidden layers.

The outputs of the ANN are 9, corresponding to the nine classes defined in Section 2.4.

The data set for training the net has been randomly divided into: 85% of samples for training, and 15% for validation. An independent data set is used for testing purposes, according to the procedure of Section 2.5.

Training uses the scaled conjugate gradient backpropagation algorithm. It automatically stops when generalization stops improving, as indicated by an increase in the cross-entropy error of the validation samples.

The following possible classifiers available in a commercial software [38] have been used for comparison purposes, which are typically used for CM applications. Among these, a preliminary selection has been carried out based on the classification accuracy. The classification accuracy is evaluated during the training phase, using 5-fold for cross-validation [39] meaning that the app partitions the data into five disjoint sets or folds and for each fold:Trains a model using the out-of-fold observations,Assesses model performance using in-fold data,Calculates the average test error over all folds.

This method gives a good estimate of the predictive accuracy of the final model trained with all the data. Feature sets have been used several times for training of both classifiers and ANNs, in repeatability post-processing conditions, and a negligible variability has been denoted in terms of accuracy of classifiers:Discriminant Classifiers, including both Linear and Quadratic Discriminant. Good classification accuracy reached in almost every group of features selected, for Linear Discriminant (accuracy >95%). It fails in almost every case when the Quadratic Discriminant classifier is used.Ensemble, including: Boosted Trees (0%, in almost every case), Subspace KNN and Bagged Trees (from 60% to 89%), Subspace Discriminant (from 72.2% up to 100%) and RUSBoosted Trees (<50%).Naïve Bayes. It fails in almost all cases with Gaussian Naïve Bayes, it ranges from 5% to 85%, with Kernel Naïve Bayes.Support Vector Machine (SVM). It performs well when Quadratic or Cubic SVM (around 90%, up to 100% in some cases), resulting in being aleatory when Linear, Fine Gaussian and Medium Gaussian SVM are chosen (from 0% to 97%).Tree, including Fine Tree, Medium Tree, and Coarse Tree. Poor accuracy of classification (around 40% maximum).k-nearest neighbor classifier (KNN), including Fine KNN, Medium KNN, Coarse KNN, Cosine KNN, Cubic KNN, and Weighted KNN. Fine KNN and Weighted KNN reach high accuracy of classification, while the behavior of others is unsatisfactory.

It has to be pointed out that all the required actions of parameters setting and optimization and data normalization have been carried out automatically by the commercial software used.

Figure 6 shows the performance of typology KNN. Looking at Figure 6 and evaluating the above-mentioned classifiers, Fine KNN has been selected as a reference for comparison with ANN, being a good option in terms of classification accuracy.

### 2.7. Performance Metrics

Different possible metrics could be used to evaluate the performance of a classifier, each of which highlights specific aspects of its behavior, so the choice among them depends on the needs of the specific application. In this CM application, whose purpose is to recognize the operating condition of the machine, in terms of jload setting and lubrication condition, we are mainly interested in the probability of detection of the state of the system, in order to restore, when necessary, the correct setting and an adequate lubrication. Moreover, we are interested in knowing the probability of false alarms, which are dangerous, because they prevent us from implementing the right intervention, or lead us to intervene incorrectly. 

For these reasons, the following metrics have been used in this work:

1. Accuracy [40,41]:(2)$$\frac{\sum_{1}^{l}\frac{tp_{i}+tn_{i}}{tp_{i}+tn_{i}+fp_{i}+fn_{i}}}{l},$$ where:
tp_i_: true positive,*fp_i_*: false positive,*fn_i_*: false negative, *tn_i_*: true negative for the class C_i_,*l*: number of classes. 


It must be considered that accuracy considers different classification errors to be equally important, adding together true positives and true negatives in the numerator of the formula [42]. For these reasons, it would be interesting to also use performance metrics that disassociate the errors that occurred in each class, such as the following two.

2. True Positive Rate (TPR), which represents the probability of detection [40,41]:(3)$$\frac{\sum_{1}^{l}tp_{i}}{\sum_{1}^{l}(tp_{i}+fn_{i})}.$$

3. False Positive Rate (FPR), which represents the probability of false alarm [40,41]:(4)$$\frac{\sum_{1}^{l}fp_{i}}{\sum_{1}^{l}(fp_{i}+tn_{i})}.$$

4. AUC (Area Under Curve), which is the area under the ROC (Receiver Operating Characteristic) graph [43], used to analyze the relationship between TPR and FPR. AUC represents the performance as a single scalar. AUC is an “ordering/rank metric”, which looks at predictions differently from the threshold metrics. If cases are ordered by predicted value, the ordering/rank metrics measure how well the ordering ranks positive cases above negative cases. The rank metrics can be viewed as a summary of the performance of a model across all possible thresholds [44].

The metrics from 1 to 3 are “threshold metrics”, that is, they depend on how the predicted values fall relative to a threshold [44]. In this work, a threshold of 0.5 has been set on the ANN output, which is a conservative and common choice [44].

## 3. Results

The proposed method of identification of a selected set of features, based on metrological and physical criteria, has been also compared with some of automatic tools, available in literature, for feature selection. The identification of the best method of selection for each specific application is out of the scope of this work. 

Considering again the results in Figure 6, it should be highlighted that the automatic tool is able to indicate a reduced number of valid features, two in this case, which are the most effective ones. It has to be pointed out that classification ability of the couple of features identified by automatic tool allows many classifiers, which have been used, to work very satisfactorily.

Figure 7 shows that the use of sets of features selected on the basis of physical and metrological criteria can be convenient (i.e., SET 1-F21 and SET 1-F16), compared to the case in which all the available features are used (i.e., SET 1-F101).

Figure 8 shows that training the ANN on a larger data set, obtained by calculating features on portions of the acquired time series, produces better results. In particular, if the 80% of the overall data set is used for training, the performance of the ANN is maximized.

Figure 9 shows a comparison between the use of the selected set by physical and metrological criteria (i.e., SET 1-F21) and couples of features suggested by an automatic procedure of selection. 

It is interesting to notice that SET 1-40 s-L10-F2 ‘linear’ and SET 1-40 s-L10-F2 ‘diaglinear’ return a different couple of selected features:-for the first case, the following features:
TD_Mean: mean values of TD,PRC98_TD: 98th percentile of TD distribution,-for the second case:
TD_Pk: range of measured values of TD,PRC98_TD: 98th percentile of TD distribution.

The meaning of features of SET 1 has been explained in [16].

Some interesting considerations arise: changing the data set to be processed, the automatic method returns different couples of features, being PRC98_TD and RMS_Scur_Mposa_diff in SET 1-20 s-L10-F2 ‘linear’, where RMS_Scur_Mposa_diff is the RMS of the differences between simulated current values (when the input of the model is the measured angular position) and Mcur. It has to be pointed out that in any case features are identified belonging to SET 1-F21.training the ANN with features on the basis of the physical and metrological method produces satisfactory results with respect to the couples of features selected by the automatic classification method.

It is interesting to observe that the score, in terms of TPR, for SET 1-F21 is in every case less than 0.89, while the AUC is equal to 1 for all classes to be classified, indicating the maximum performance. This means that the values corresponding to positives are the highest for each class, and this is a further point in favor of the classification method.

Figure 10 presents a summary of the results for classification when the method for feature selection, the data set characteristics, and the number of features are changed. Results for ANN and the fine KNN algorithm, previously acknowledged as the best one among the all tested classifiers, are shown: it can be seen that in most cases the ANN and classifier performance is very satisfactory; in particular, with SET1-20 s-L5-F2 ‘linear’, SET1-40 s-L10-F16, SET1-20 s-L5-F21, and SET1-40 s-L10-F21, both the ANN and the classifier work well. This confirms the goodness of the selection method by physical and metrological criteria; in fact, in any case, the selected features are according to this criterion. It has to be noted that the features of SET1-20 s-L5-F2 ‘linear’, automatically selected, belong to the set of 21 or 16 features by the physical and metrological approach.

Instead, it can be seen that neither of the two considered methods satisfactorily classifies when SET 2 is used.

Moreover, the ANN works better when the training data set is large. This confirms the observation made with reference to Figure 8.

As a result of further tests, the addition of noise, for enlarging the variability of the training set, does not produce improvements on the ANN performance.

## 4. Conclusions

In this paper, a physical and metrological approach is presented for extracting and selecting features for CM applications, with reference to an automatic mechatronic system in real scale for packaging applications.

The described approach is hybrid, that is, the initial feature set has been built on the basis of experimental data from sensors and of results supplied by a simplified kinematic and dynamic model of the system. This approach has suggested the features that could be most sensitive, from a physical point of view, to changes in jload setting and lubrication conditions. Repeatability of features is a further criterion of selection from a metrological point of view.

Then, the most significant features among those identified have been selected, to be processed by advanced classification algorithms, based on ANN or other classifiers.

The effect of some aspects of the procedure on the classification performance has been analyzed, like: the choice of the temporal window for the feature calculation, the choice of different groups of features, and the addition of noise to the data. Results about a comparison with the state-of-the-art of automatic feature extraction and selection have also been discussed.

If the features set selected by the physical and metrological criterion is used, a high score in terms of TPR is achieved, classifying by ANN different setting and lubrication conditions. Furthermore, corresponding AUC values equal to 1 are found, confirming a reliable capability of classification. This result also confirms that the initial set of features is itself promising.

If the use of the ANN is taken into account, the main results are according to the following:improvements due to added noise to the data are negligible;the acquisition length being fixed, ANN benefits from having more temporal windows for supervised classification.

About the comparison of the proposed method with automatic ones, the main indications of this experimental work show that:
a group of very few features can be found for classification purposes;the features automatically selected belong to the group suggested by metrological and physical criteria;if the tests on which the features are calculated are changed, the selected features are partially changed too, even though they are still in the group suggested by metrological and physical criteria;ANN used for classification does not work with automatically selected features, which operate satisfactorily only with some classifiers (e.g., Fine KNN and Weighted KNN).

In future work, the analysis will be deepened using a higher number of test and training samples.

Further experimental analysis for the identification of the best criteria to be used for automatic feature extraction will be carried out, for a more exhaustive comparison.

## Figures and Tables

**Figure 1 sensors-19-05186-f001:**
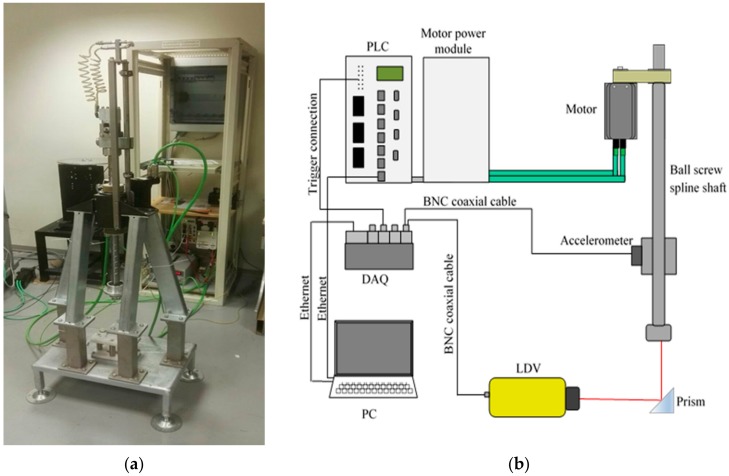
Experimental test bench: (**a**) picture [11]; (**b**) scheme of the measurement and data acquisition apparatus.

**Figure 2 sensors-19-05186-f002:**
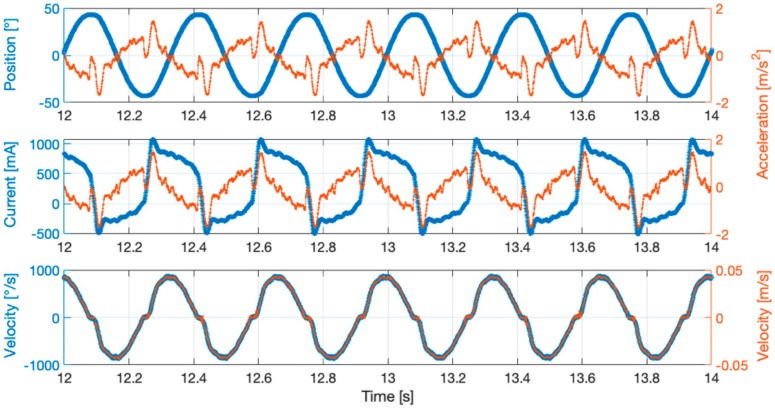
Example of experimental time histories. Respectively: angular position (°) and linear acceleration (m/s^2^); motor current (mA) and linear acceleration (m/s^2^); angular velocity (°/s) and linear velocity (m/s).

**Figure 3 sensors-19-05186-f003:**
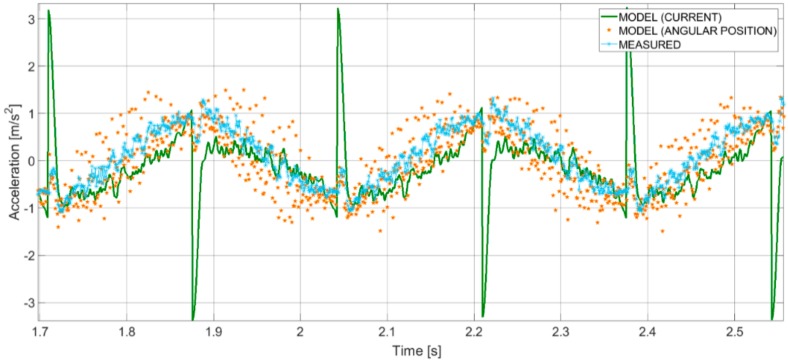
Comparison between simulated and measured acceleration signals [17]. MODEL (CURRENT) is the acceleration simulated, using as an input of the model the measured current; MODEL (ANGULAR POSITION) is the acceleration simulated, using as an input of the model the angular position. MEASURED is the measured acceleration signal.

**Figure 4 sensors-19-05186-f004:**
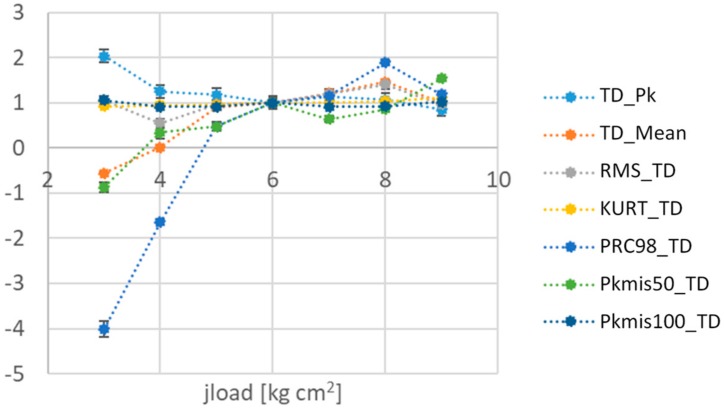
Trends of the features calculated on the basis of Tracking Deviation (TD), at different jload [22]. Repeatability is shown as error bars.

**Figure 5 sensors-19-05186-f005:**
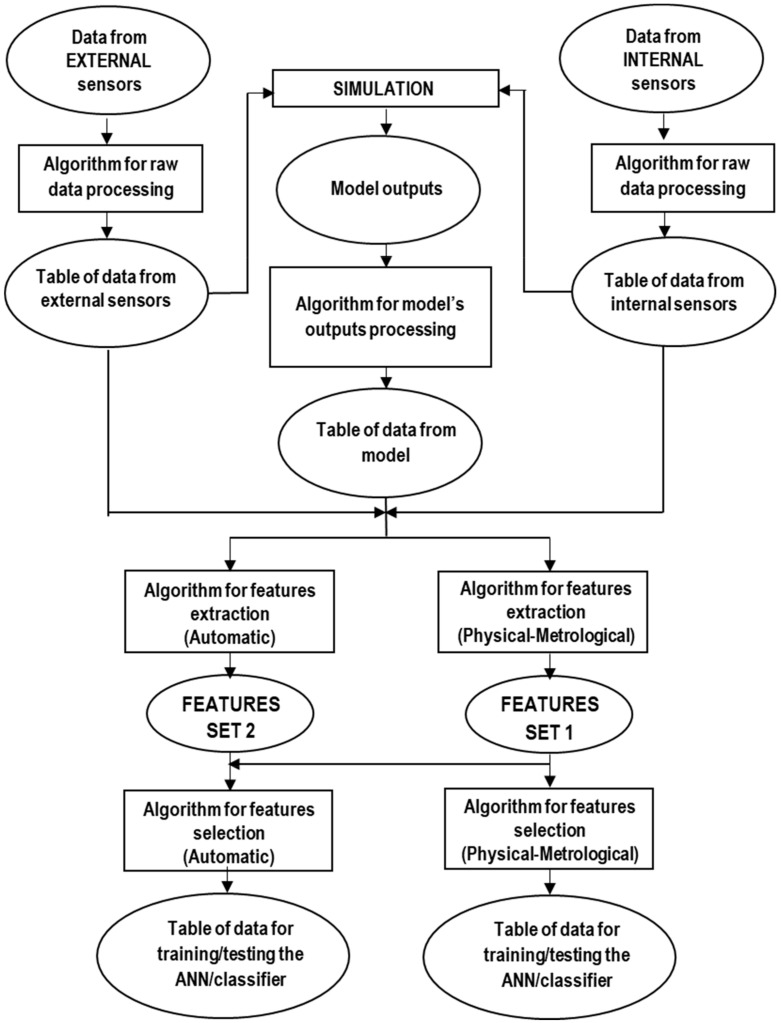
Flow diagram of the main steps of data processing.

**Figure 6 sensors-19-05186-f006:**
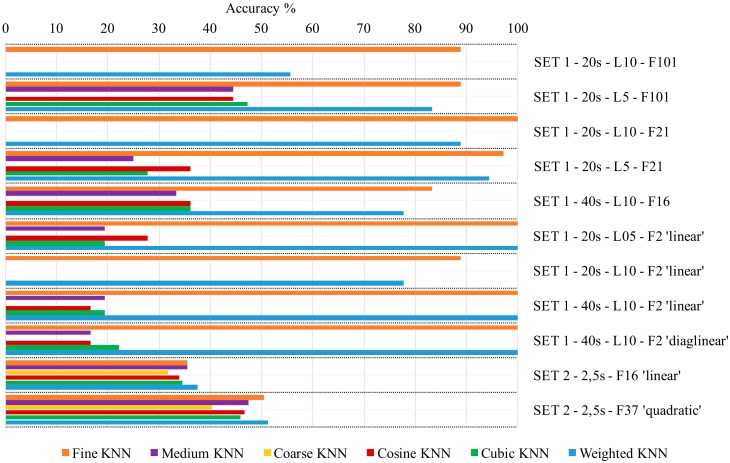
**K**-nearest neighbor classifiers: comparison among different configurations.

**Figure 7 sensors-19-05186-f007:**
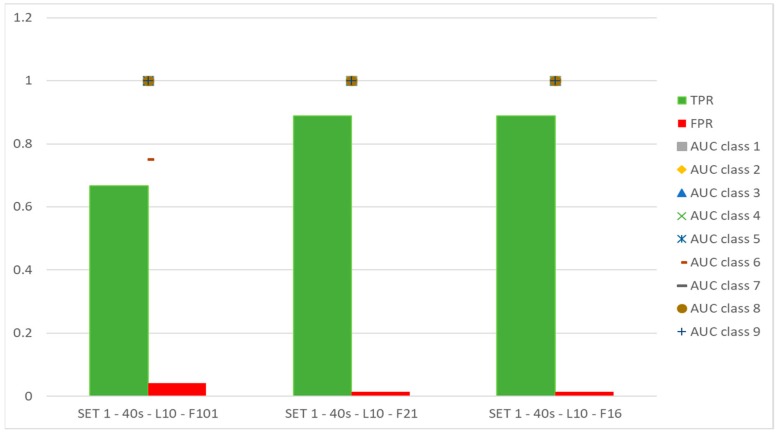
Comparison between all features and groups of features selected by the physical and metrological method. Artificial Neural Network is used for classification.

**Figure 8 sensors-19-05186-f008:**
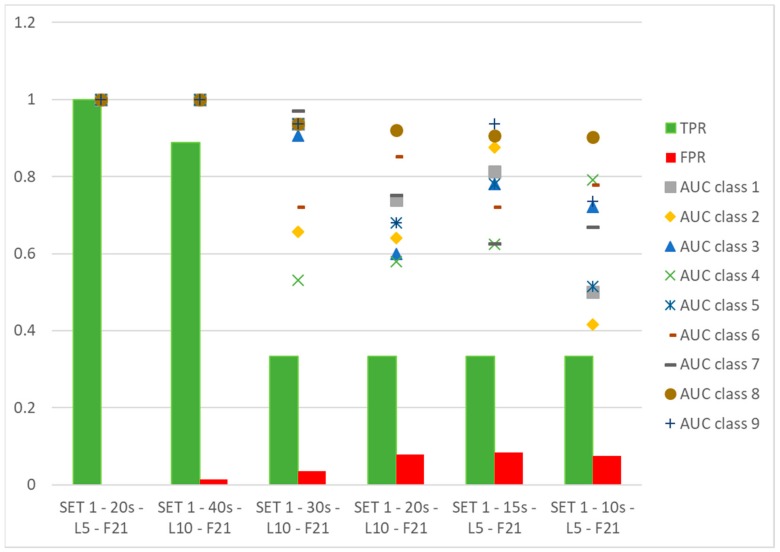
Comparison among cases corresponding to different time bases for the features calculation. ANN is used for classification.

**Figure 9 sensors-19-05186-f009:**
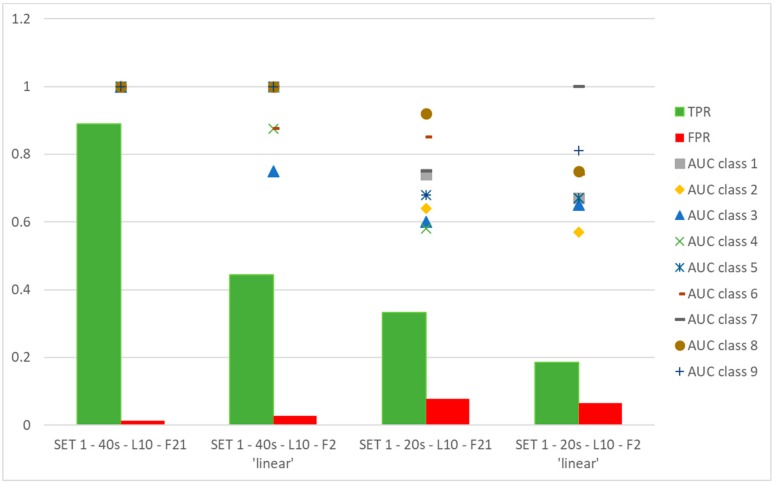
Comparison among automatic and physical-metrological selection methods. ANN is used for classification.

**Figure 10 sensors-19-05186-f010:**
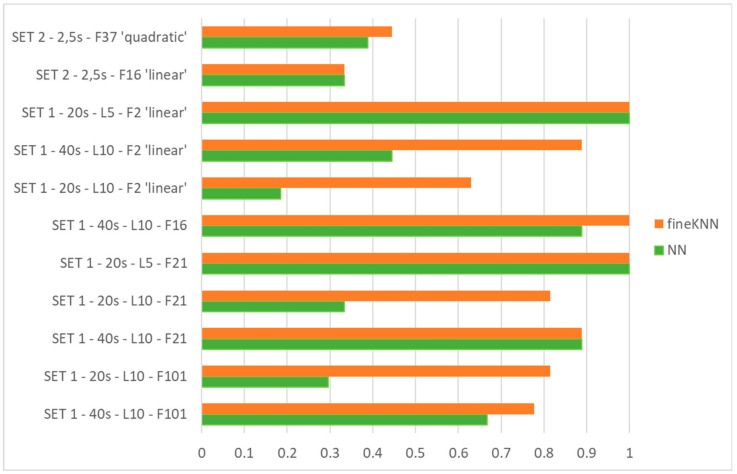
Comparison between performances of ANN and fine KNN (performance indicator: True Positive Rate).

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
