# Peer review of "Physical and Metrological Approach for Feature’s Definition and Selection in Condition Monitoring†"

_sensors, 2019, doi:10.3390/s19235186_

Round 1
Reviewer 1 Report
Starting with the abstract, some sentences are not complete or otherwise not structured, e.g. “This interaction suggests the features that could be mostly correlated, from the physical point of view, to the status of the system to be identified.” The abstract is not clear when it refers to a mechatronic system – it is difficult to understand what does this paper talks about. Line 32 presents references that are weak examples, also reference 5 in line 47 is very weak – there are many stronger examples in the literature.
This paper is about condition monitoring but still half way through the paper the reader does not know what is the “condition” being monitored! – even though the apparatus has been described later on! In Section 2.3 regarding feature selection there is still no reference to the condition being monitored or otherwise related indicators to the condition! Strangely feature selection methodology is referenced through a link to a Mathwork´s procedure – no authors or papers are referenced, there is no method of reference. Figure 3 is incomplete (right upper corner). Finally in line 148 there is a reference to the condition being monitored!
The paper presents many and different machine learning approaches but none of them is clearly presented, regarding their configuration. For example, when using k-NN one does not know the distance measure used; what is the stop criteria; how many classes or clusters are identified! This is just one of the classifiers, not to mention the others!
The paper refers ANN but it does not mention the type of ANN, if a simple feedforward network, a SOM, a ART, RNN, a Convolutional …. !!!?? No training specs are provided, no feature vector is provided, no training methodology is given!
The paper presents a real problem with many potential applications but the methods that sustain the results are not clear or otherwise explained properly.
Author Response
Starting with the abstract, some sentences are not complete or otherwise not structured, e.g. “This interaction suggests the features that could be mostly correlated, from the physical point of view, to the status of the system to be identified.” The abstract is not clear when it refers to a mechatronic system – it is difficult to understand what does this paper talks about. Line 32 presents references that are weak examples, also reference 5 in line 47 is very weak – there are many stronger examples in the literature.
The abstract has been revised with reference to the Reviewer’s notes. In the introduction, references have been added and discussed in order to give a wider point of view of the state of the art.
This paper is about condition monitoring but still half way through the paper the reader does not know what is the “condition” being monitored! – even though the apparatus has been described later on! In Section 2.3 regarding feature selection there is still no reference to the condition being monitored or otherwise related indicators to the condition!
The condition being monitored have been cited in the abstract, in the introduction and in a more detailed way in Section 2.4 (starting from former line 198).
Strangely feature selection methodology is referenced through a link to a Mathwork´s procedure – no authors or papers are referenced, there is no method of reference.
References have been added. Even though many new possible approaches for feature selection can be used, which are typically applied in specific cases, the proposed method was intentionally compared with a method of large utilization and easy application, validated also in many cases, in order to give a comparison of more general validity.
Figure 3 is incomplete (right upper corner).
The figure has been redrawn in order to insert the missing information and to improve understandability.
Finally in line 148 there is a reference to the condition being monitored!
The condition being monitored have been cited in the abstract, in the introduction and in a more detailed way in Section 2.4 (starting from former line 198).
The paper presents many and different machine learning approaches but none of them is clearly presented, regarding their configuration. For example, when using k-NN one does not know the distance measure used; what is the stop criteria; how many classes or clusters are identified! This is just one of the classifiers, not to mention the others!
The paper refers ANN but it does not mention the type of ANN, if a simple feedforward network, a SOM, a ART, RNN, a Convolutional …. !!!?? No training specs are provided, no feature vector is provided, no training methodology is given!
The paper presents a real problem with many potential applications but the methods that sustain the results are not clear or otherwise explained properly.
Section 2.6 concerning the different machine learning approaches taken into account has been reorganized. ANN type, training specification, training methodology are described. The following sentence has been added to the text: It has to be pointed out that all the required actions of parameters setting and optimization and data normalization have been carried out automatically by the used commercial software.
Reviewer 2 Report
The authors give a good overview of different data-based approaches to condition monitoring, describe their mechanical setup for training data generation, their approach to hybrid feature extraction including both measured and simulated data. The authors proceed to select the best features for load and lubrication state classification and test the selected features with multiple, different classifiers.
Despite the good introduction the explanation of the employed algorithms for feature extraction and selection and the evaluation of these methods need substantial improvements. Also the language needs to be improved. The following is a list of necessary improvements, problems to be solved and open questions. It also includes some (not by far all) language improvements.
General: The abstract claims core of the paper is the hybrid approach to feature extraction and the physical and metrological selection of features. However none of the algorithms is sufficiently explained. E.g. the explanation of the metrological approach for feature selection spans only the lines 178 and 179. Scientific standards require the authors to explain everything an informed reader needs to achieve the same results as the authors by employing the explanations in the paper. For that the explanations need to be far more elaborate. General: Chapter 2.2 introduces two datasets, however the results in chapter 3 are only shown for set 1. Therefore there is no comparison of the proposed method to any comparable method for feature extraction. Design of experiment: If the ration j/k that ranges from 1-8 defines the number of observations used for training the design of experiment completely ignores the statistical nature of the machine learning algorithms used. Why was the considered timespan as low as 40s when measurements were taken by a seemingly automated test setup? The considered total length of acquisition should be way longer. Chapter 2.6: Please give an explanation for so many algorithms failing to get any results or results that are worse than random guessing. Chapter 2.7: This chapter should be reduced to the explanation of the actually used performance metrics. Chapter 3: The authors almost exclusively show results achieved using an ANN classifier. However figure 8 shows equal or better performance by fine KNN classifier in all tested scenarios. Neither the reason for choosing ANN nor its architecture and hyperparameters are named. Again the explanations should be detailed enough to enable the reader to achieve the same results as the authors, which is not the case. Line 105: Figure 1b instead of figure 2b Line 109: Sampling rates and bandwidths of the sensors should be mentioned Line 117-136: The enumeration has far wider linespaces that any other enumeration in the paper. They should be reduced. Figure2: The right axis label is cut off. Line 123: Multiple runs of the model instead of Multiple running of the model Line 134: Explain jload Line 137-140: Correct this sentence Line 146: A careful analysis of the accuracy of features has been carried out. Please reference, where this analysis can be found. Line 156: How are the hybrid features extracted? Please explain in detail. Line 160: [23-26] instead of [23], [24-26] Line 174: How is the kinematic model asked to indicate meaningful features? Please describe the algorithm in detail. Line 178: Please describe the used algorithm in detailed. Line 180/181: …, with respect to other, … Part of the sentence seems missing Line 186-188: Which of the >20 approaches named in reference 27 has been employed. Why exactly this approach? Line 190/191: This sentence seems to make no sense. Please check. Figure 3: Please add Set1 and Set2 for better understanding. Figure 3: No algorithms for raw data preprocessing were mentioned or explained. Figure 3: Please explain “Outpputs calculation from different inputs” in the figure caption? Figure 3: Data from internal sensors instead of data from internal Figure 3: Is there any feature preprocessing, e.g. standardization as required by SVM and KNN applied before classifier training? Line 202: Please explain why this specific condition was chosen for test data acquisition? Line 223-226: Were time windows synchronized with the machine`s movements? If not all features sensitive to temporal shifts should be excluded from analysis. Line 229/Line 232: derived instead of deriving Line 236: If the Matlab function sequentialfs was used the feature selection approach employed was commercially available in 2008 and does not reflect the state of science due to the huge scientific efforts in this field. Chapter 2.6: Usage of the classification app should also be mentioned in the text. Line 283: Please explain which value is being thresholded. Line 290: Is the case of very different prior probabilities relevant in this paper? Line 327: Please reference the results that show that. Line 338: Figure 7 does not show Set1-F16 Line 345: Please describe the architecture of the ANN and the values of the hyperparameters used. Line 353-357: Please define the abbreviations before their usage Line 358: less instead of more Line 360-361: Please elaborate this reasoning. If AUC == 1 there exists a threshold with TPR == 1 and FPR == 0. Why is this threshold not used? Line 362: and instead of e Line 364: KNN is better or equal to ANN in every shown case Line 366: Why do authors believe that additional noise to an already poor representation of the data distribution would benefit the classifier? As well as the low number of training examples this completely contradicts the statistical nature of machine learning algorithms. Figure 5: Please name the classifier used. Please describe classes 1-9 (also for figures 6 and 7). Why are the results of shown for 40s-L10? What happens in other cases? Line 391: given the small number of training/testing examples and the huge number of failing classifiers the reliability of the classification seem in doubt. Line 406: Please name an example for an optimized class of classifiers for better understanding. Line 408: Analysis with an higher number of test and training samples should also be part of the outlook. References: All references of websites should state the date accessed.Author Response
The authors give a good overview of different data-based approaches to condition monitoring, describe their mechanical setup for training data generation, their approach to hybrid feature extraction including both measured and simulated data. The authors proceed to select the best features for load and lubrication state classification and test the selected features with multiple, different classifiers.
Despite the good introduction the explanation of the employed algorithms for feature extraction and selection and the evaluation of these methods need substantial improvements. Also the language needs to be improved. The following is a list of necessary improvements, problems to be solved and open questions. It also includes some (not by far all) language improvements.
General: The abstract claims core of the paper is the hybrid approach to feature extraction and the physical and metrological selection of features. However none of the algorithms is sufficiently explained. E.g. the explanation of the metrological approach for feature selection spans only the lines 178 and 179. Scientific standards require the authors to explain everything an informed reader needs to achieve the same results as the authors by employing the explanations in the paper. For that the explanations need to be far more elaborate.
The hybrid approach to feature extraction has been better explained in Section 2.1 and by means of Figure 3.
The physical and metrological selection of features has been better explained inn Section 2.3 and by means of Figure 4
General: Chapter 2.2 introduces two datasets, however the results in chapter 3 are only shown for set 1. Therefore there is no comparison of the proposed method to any comparable method for feature extraction.
Results have been added in Figure 10 and commented.
Design of experiment: If the ration j/k that ranges from 1-8 defines the number of observations used for training the design of experiment completely ignores the statistical nature of the machine learning algorithms used. Why was the considered timespan as low as 40s when measurements were taken by a seemingly automated test setup? The considered total length of acquisition should be way longer.
The time duration of acquisition has been set with reference to a range including a minimum number of operation cycles and a maximum one related to the need of acquiring too many samples. 40 s has been considered a right trade-off.
This sentence has been added to the text (Section 2.5).
Chapter 2.6: Please give an explanation for so many algorithms failing to get any results or results that are worse than random guessing.
Selecting the right status among the considered ones is not trivial, being the differences among the conditions to be classified not very remarkable. Furthermore, the authors considered useful to highlight that only a part of classifiers of general use is able to recognize the right statuses.
Chapter 2.7: This chapter should be reduced to the explanation of the actually used performance metrics.
The chapter has been reduced, as suggested.
Chapter 3: The authors almost exclusively show results achieved using an ANN classifier. However figure 8 shows equal or better performance by fine KNN classifier in all tested scenarios.
Comments about results of Figure 8 (in the new version Figure 10) have been completely rewritten, in order to present the goal of the comparison, which is not really linked to show which classifier is better; in facts, the aim is to confirm the goodness of the selection method by physical and metrological criteria.
Neither the reason for choosing ANN nor its architecture and hyperparameters are named. Again the explanations should be detailed enough to enable the reader to achieve the same results as the authors, which is not the case.
The requested information about the ANN has been added in Section 2.6.
Line 105: Figure 1b instead of figure 2b
Done
Line 109: Sampling rates and bandwidths of the sensors should be mentioned
Done
Line 117-136: The enumeration has far wider linespaces that any other enumeration in the paper. They should be reduced.
Done
Figure2: The right axis label is cut off.
Done. Figure caption has been extended to improve understandability.
Line 123: Multiple runs of the model instead of Multiple running of the model
Done
Line 134: Explain jload
Done. The following sentence has been added into the text: Jload is a parameter of the control system, representing the load inertia at the motor axis.
Line 137-140: Correct this sentence
Done. The sentence has been modified and extended.
Line 146: A careful analysis of the accuracy of features has been carried out. Please reference, where this analysis can be found.
The sentence has been modified, including the reference to the related section (Section 2.3).
Line 156: How are the hybrid features extracted? Please explain in detail.
The hybrid features definition has been better explained in Section 2.1 and by means of Figure 3
Line 160: [23-26] instead of [23], [24-26]
Done
Line 174: How is the kinematic model asked to indicate meaningful features? Please describe the algorithm in detail.
This sentence has been moved to Section 2.2, related to the definition of the initial set of features. At the moment, there is not an algorithm for the automatic indication of features. The definition of meaningful features has been carried out according to the criteria described in the paper in Section 2.1.In particular, the hybrid features definition has been better explained by means of Figure 3.
Line 178: Please describe the used algorithm in detailed.
The metrological selection of features has been better explained In Section 2.3 by means of Figure 4. Although there is not an algorithm, this procedure could be easily automated. At the moment, the selection has been carried out by means the analysis of graphs like those of figure 4.
Line 180/181: …, with respect to other, … Part of the sentence seems missing
This sentence has been modified.
Line 186-188: Which of the >20 approaches named in reference 27 has been employed. Why exactly this approach?
Many possible approaches for feature selection could be used, among them, the Matlab function sequentialfs was used, being of large utilization and easy application.
Line 190/191: This sentence seems to make no sense. Please check.
The sentence has been deleted. No useful information is given.
Figure 3: Please add Set1 and Set2 for better understanding.
Done. Please note that Figure 3 is now Figure 5.
Figure 3: No algorithms for raw data preprocessing were mentioned or explained.
Explanation is given in the text in Section 2.3. The algorithms for raw data preprocessing have been extensively described in [38].
Figure 3: Please explain “Outpputs calculation from different inputs” in the figure caption?
Figure 3: Data from internal sensors instead of data from internal
The expression “Outputs calculation from different inputs” refers to step 3 of the methodology in section 2.1. In any case, the sentence has been eliminated, being unclear and even redundant, since from the scheme it is evident that different types of inputs (from internal and external sensors) enter the model, and outputs come out correspondingly.
Figure 3: Is there any feature preprocessing, e.g. standardization as required by SVM and KNN applied before classifier training?
The following sentence has been added to the text (Section 2.6): It has to be pointed out that all the required actions of parameters setting and optimization and data normalization have been carried out automatically by the used commercial software.
Line 202: Please explain why this specific condition was chosen for test data acquisition?
The test condition was chosen because it is close to the maximum rate of the system (Section 2.4)
Line 223-226: Were time windows synchronized with the machine`s movements? If not all features sensitive to temporal shifts should be excluded from analysis.
A part of features is evaluated considering only a part of the operating cycle and for these, time windows are synchronized with the machine’s movements (e.g. Figure 3). The remaining part of features are evaluated considering an integer number of operating cycles (e.g. RMS, kurtosis, features in the frequency domain, …).
Line 229/Line 232: derived instead of deriving
Done
Line 236: If the Matlab function sequentialfs was used the feature selection approach employed was commercially available in 2008 and does not reflect the state of science due to the huge scientific efforts in this field.
We agree that many new possible approaches for feature selection can be used. The proposed method was intentionally compared with a method of large utilization and easy application, validated also in many cases, in order to give a comparison of more general validity.
Chapter 2.6: Usage of the classification app should also be mentioned in the text.
A reference to the classification learner app has been added.
Line 283: Please explain which value is being thresholded.
The thresholded value is the ANN output (Section 2.7).
Line 290: Is the case of very different prior probabilities relevant in this paper?
This is not the case of very different prior probabilities, and the sentence in former line 290 has been removed.
Line 327: Please reference the results that show that.
The results in Figure 6 show that. This has been pointed out in the text (Section 3).
Line 338: Figure 7 does not show Set1-F16
Corrected. The reference to SET1-F16 has been removed from the text.
Line 345: Please describe the architecture of the ANN and the values of the hyperparameters used.
In Section 2.6 ANN type, training specification, training methodology are described.
Line 353-357: Please define the abbreviations before their usage
The abbreviations have been defined when used.
Line 358: less instead of more
Corrected
Line 360-361: Please elaborate this reasoning. If AUC == 1 there exists a threshold with TPR == 1 and FPR == 0. Why is this threshold not used?
We agree and this a good suggestion. However, we set a fixed threshold in order to consider the effect of random variability of the ANN’s results.
Line 362: and instead of e
Corrected
Line 364: KNN is better or equal to ANN in every shown case
This concept has been better explained.
Line 366: Why do authors believe that additional noise to an already poor representation of the data distribution would benefit the classifier? As well as the low number of training examples this completely contradicts the statistical nature of machine learning algorithms.
We considered this observation. In the text the sentence has been modified referring to variability of the training set. In some cases of machine learning applications, insertion of limited level of noise improve the classification capability of ANNs (Giulio D'Emilia, David Di Gasbarro, Antonella Gaspari, Emanuela Natale, “Managing the uncertainty of conformity assessment in environmental testing by machine learning”, Measurement 124, 2018, pp. 560–567; D'Emilia, G., Marra, A., Natale, E., “Use of neural networks for quick and accurate auto-tuning of PID controller”, Robotics and Computer-Integrated Manufacturing, 23-2, 2007, pp. 170-179). This is not the case, mainly due to the limited number of training samples.
Figure 5: Please name the classifier used.
ANN is used for classification, and this has been pointed out in the caption.
Please describe classes 1-9 (also for figures 6 and 7).
The 9 classes are obtained by combining 3 lubrication conditions and 3 jload values (3 x 3 = 9). This has been pointed out in section 2.4.
Why are the results of shown for 40s-L10? What happens in other cases?
Former figure 5 aimed at showing the behavior of different groups of features being fixed other conditions. The indications gained in other cases are similar.
Line 391: given the small number of training/testing examples and the huge number of failing classifiers the reliability of the classification seem in doubt.
Despite the small number of training/testing examples, a part of classifiers of general use is able to recognize the right statuses, and this, in the opinion of the authors, confirm that the initial set of features is promising. In future work the analysis will be deepened using a larger data set.
Line 406: Please name an example for an optimized class of classifiers for better understanding.
The sentence has been modified and an example of working classifier is given.
Line 408: Analysis with an higher number of test and training samples should also be part of the outlook.
We agree with the reviewer, and this outlook has been added to the Conclusions section.
References: All references of websites should state the date accessed.
Done
Reviewer 3 Report
Comments
===========
1) Section 2.2 (Definition of the initial set of futures) has been written to general. Most important issues are not clarified but only references to 15 publications are given. This chapter needs improvement and a more detailed clarification of issues such as:
- the process of building the initial group of features,
- the conception of hybrid futures,
- which commercially available features extraction methods and how were they chosen?
- the idea of separating futures into two separate sets (SET1, SET2), with one having 101 futures and the other ... 100.
2) The same remark applies to chapter 2.3. The way the features are selected and the procedure of ANN training/testing is carried out should be further explained.
Editorial remarks
==============
1) Line 105: Figure 1(b) instead 2(b)
2) Figure 2: Right y-axis labels are cropped, thus illegible. The left y-axis label on the third plot should be described more precisely as "angular velocity".
3) Figure 3: "Data from INTERNAL sensors" instead "Data from INTERNAL"
Author Response
Comments
===========
1) Section 2.2 (Definition of the initial set of futures) has been written to general. Most important issues are not clarified but only references to 15 publications are given.
This chapter needs improvement and a more detailed clarification of issues such as:
- the process of building the initial group of features,
- the conception of hybrid futures,
A part of selected features are the most diffused in literature. The novelty is in the application both to real and simulated data. The hybrid features definition has been better explained, also with reference to section 2.1 and Figure 3.
- which commercially available features extraction methods and how were they chosen?
Many possible approaches for feature extraction can be used. The proposed method was intentionally compared with a method of large utilization and easy application, in order to give a comparison with an approach that does not require physical knowledge of the application. This sentence has been inserted into the text.
- the idea of separating futures into two separate sets (SET1, SET2), with one having 101 futures and the other ... 100.
The two sets used for comparison have been defined approximately of the same size in order to avoid the effect of the size set.
2) The same remark applies to chapter 2.3. The way the features are selected and the procedure of ANN training/testing is carried out should be further explained.
The physical and metrological selection of features has been better explained in Section 2.3 and by means of Figure 4.
Section 2.6 concerning the different machine learning approaches taken into account has been reorganized. ANN type, training specification, training methodology are described.
Editorial remarks
==============
1) Line 105: Figure 1(b) instead 2(b)
Corrected
2) Figure 2: Right y-axis labels are cropped, thus illegible. The left y-axis label on the third plot should be described more precisely as "angular velocity".
The figure is now legible. A more detailed caption has been added.
3) Figure 3: "Data from INTERNAL sensors" instead "Data from INTERNAL"
The figure has been redrawn in order to insert the missing information and to improve understandability.
Round 2
Reviewer 1 Report
The suggested amendments were in general attended. The ANN information introduced suffices and provides adequate support for further considerations.
Reference numbering does not seem to follow a coherent pattern, i.e. for example, in line 38 consecutive references appear different from those in line 75. Should be reviewed – attend to the journal’s standard.
Author Response
Thank you for the suggestion. Reference numbering has been corrected according to the journal’s standard.
Reviewer 2 Report
The authors implemented all key suggestions for improvements.
There are still some minor spelling/gramar mistakes left that could be improved.
Author Response
A text review has been carried out; some sentences have been made more simple and many errors have been corrected, according to the new suggestions of the Reviewer (see for instance rows 78, 174, 191, 247, 278, 287, 383, 397 of the revised paper).
Reviewer 3 Report
Dear Authors, thank you for your comprehensive response to all my questions and remarks.
Author Response
The authors thank you for your contribution to the improvement of the technical content of the paper.